# Navigating Ambivalence: Artificial Intelligence and Its Impact on Student Engagement in Engineering Education

**DOI:** 10.3390/bs16010011

**Published:** 2025-12-20

**Authors:** Liliana Pedraja-Rejas, Patricio Lazo Vega, Pablo Rojas Huanca

**Affiliations:** Departamento de Ingeniería Industrial y de Sistemas, Universidad de Tarapacá, Casilla 7D, Arica 1020000, Chile; patricio.lazov10@gmail.com (P.L.V.); pablorojas.huanca@gmail.com (P.R.H.)

**Keywords:** artificial intelligence (AI), technology-enhanced learning, digital pedagogy, technology acceptance, student engagement, gender differences, higher education, Latin America

## Abstract

Artificial intelligence (AI) is rapidly transforming higher education, yet limited empirical evidence exists on how students experience the emotional, cognitive, and ethical tensions associated with AI, particularly in Latin American contexts. This study addresses this gap by examining patterns of adoption, perceived usefulness, and the ambivalent experiences that arise when engaging with AI tools for academic learning. A questionnaire combining closed and open-ended questions was administered to 170 engineering students from a Chilean public university. A mixed-methods design was used to analyse the data: quantitative analyses identified adoption patterns and perceived usefulness, while qualitative thematic analysis captured emerging emotional, ethical, and motivational tensions. The results showed high adoption (73.5%), driven by the pragmatic usefulness of saving time, understanding concepts, and improving work. Although the overall perception was positive, a deep ambivalence was identified, with enthusiasm and confidence coexisting with ethical (plagiarism), cognitive (dependence), and technical (reliability) concerns. It is concluded that the effective integration of AI transcends technological access and requires an institutional strategy that promotes critical digital literacy, clear policies, and support programmes that address competency and gender gaps, ensuring ethical and equitable adoption that enhances learning without compromising the development of critical thinking.

## 1. Introduction

Artificial intelligence (AI) has established itself as one of the most impactful technological innovations in contemporary education. In higher education, these tools offer opportunities to optimise teaching and learning processes, personalise training, and prepare students for a highly digitised and competitive job market ([39]). However, incorporating AI into educational environments remains a complex challenge. Resistance to change, limited understanding of its potential, and persistent concerns, ranging from fears of replacing human skills to doubts about the accuracy and credibility of AI outputs, continue to hinder adoption ([5]; [26]). These barriers are often exacerbated in Latin American contexts, where uneven digital infrastructures and gaps in institutional readiness restrict effective implementation, underscoring the need for locally grounded empirical evidence to inform equitable and context-sensitive integration strategies ([33]).

In this scenario, AI literacy has emerged as a critical competency that extends far beyond operational or instrumental skills. It involves the ability to critically interrogate AI tools, understand their affordances and constraints, and use them autonomously, responsibly, and ethically. Recent studies show that students with higher levels of AI literacy perceive greater usefulness, display more favourable adoption intentions, and demonstrate stronger self-regulation in learning ([1]; [39]).

At the same time, student engagement has become a key construct for assessing the actual impact of technology on learning. Defined as the cognitive, emotional, and behavioural involvement of students in their educational process, engagement has been consistently associated with higher levels of motivation, self-regulation, and academic performance ([16]; [20]). Emerging evidence indicates that educational technologies, when embedded within coherent pedagogical frameworks, can strengthen engagement by providing interactive learning environments, personalised guidance, timely feedback, and adaptive forms of support that foster active and sustained participation ([4]; [14]; [23]). In this sense, technology’s capacity to scaffold learning processes and support autonomy is regarded as one of its most distinctive contributions to meaningful and durable engagement.

However, this potential is not uniformly realised. Growing evidence suggests that students’ engagement with AI tools is shaped by a constellation of psychological, contextual, and socio-emotional factors, not solely by technical performance. Key determinants include students’ perceived competence, their trust in AI systems, their emotional comfort when interacting with such tools, and the extent to which learning environments support autonomy and critical agency ([41]; [43]). When these factors misalign, an emerging “AI engagement gap” can be observed: a divergence between the availability of AI tools and students’ actual capacity, willingness, or confidence to use them meaningfully. This gap often manifests as uneven or inconsistent adoption, surface-level or short-term use, avoidance behaviours, and elevated feelings of anxiety or mistrust. Such patterns have been widely documented in research on gendered engagement with technology and STEM learning environments, where women tend to report lower self-efficacy, higher technological anxiety, and reduced sense of belonging ([3]; [8]; [13]).

Despite the rapid expansion of AI in higher education, empirical evidence remains limited regarding how students navigate these cognitive, emotional, and ethical tensions in everyday academic practice, particularly in Latin American contexts, where structural, cultural, and institutional dynamics may shape experiences differently. Understanding how engineering students, who are often early adopters of digital tools, interpret the benefits, risks, and emotional ambiguities of AI is therefore essential for designing pedagogical strategies that not only introduce the technology but also cultivate critical, responsible, and equitable engagement.

In light of these considerations, the present study seeks to offer an integrated understanding of engineering students’ experiences with AI by examining their patterns of use, their perceptions of usefulness, and the emotional and ethical tensions that arise when engaging with these tools. To guide this analysis, the study addresses the following research questions:

RQ1. What are the patterns of use and adoption of AI tools among engineering students, and what motivations underlie their usage?

RQ2. How do students perceive the usefulness and impact of AI tools on their learning, particularly in relation to efficiency, autonomy, and conceptual understanding, and are there gender-based differences in these perceptions?

RQ3. What benefits, risks, difficulties, and emotions emerge from students’ experiences when using AI tools in academic contexts?

By answering these questions, the study seeks to contribute to contemporary debates on technology-mediated learning and to support the development of ethical, inclusive, and student-centred approaches to the adoption of AI in higher education.

## 2. Theoretical Framework

### 2.1. AI in Higher Education: Promise, Limitations, and Emerging Tensions

AI has become a central component of contemporary higher education, offering opportunities for personalisation, adaptive support, and efficient access to information ([12]; [39]). In engineering education specifically, AI-powered systems facilitate problem-solving, modelling, and conceptual understanding through simulations and data-driven feedback ([9]; [32]; [34]). These affordances position AI as a pedagogical ally capable of strengthening cognitive and procedural learning.

However, its integration is accompanied by persistent ethical, emotional, and functional tensions. Concerns regarding accuracy, opacity, and the risk of cognitive dependence coexist with enthusiasm for efficiency and personalised support. Students also express apprehension about plagiarism, unclear institutional policies, and the potential misalignment between AI-generated content and academic integrity frameworks ([10]; [11]; [14]; [17]; [25]; [26]; [28]). These tensions are heightened in contexts with uneven digital infrastructures or limited institutional readiness, such as many Latin American universities ([33]; [38]).

A growing body of research conceptualises these mixed evaluations through the psychological construct of attitudinal ambivalence, defined as the simultaneous coexistence of positive and negative evaluations toward the same object ([22]). In the case of AI, ambivalence manifests as attraction to its pedagogical benefits coupled with mistrust, anxiety, or ethical concern. Recognising this ambivalence is essential, as it shapes adoption patterns, depth of engagement, and the extent to which AI is used meaningfully rather than superficially.

### 2.2. AI-Mediated Learning and the Role of Autonomy, Competence, and Feedback

AI-based learning environments provide personalised recommendations, adaptive difficulty, and real-time explanations that support students’ capacity to learn autonomously and regulate their progress ([5]; [14]). These systems align with engineering students’ need for independent problem-solving and the development of technical reasoning ([32]). Recent studies show that AI can act as a continuous “learning companion” that offers step-by-step guidance, clarification of complex concepts, and iterative improvement of written work ([21]; [36]; [42]).

The motivational effectiveness of these AI–student interactions can be robustly analysed through the lens of Self-Determination Theory (SDT) ([15]; [30]). SDT posits that sustained engagement emerges when educational experiences satisfy three fundamental psychological needs: autonomy, competence, and relatedness.

Autonomy is strengthened when students can explore content independently, self-regulate their learning pace, and access immediate assistance when needed. Recent research in higher education has documented that artificial intelligence tools, such as conversational assistants, foster precisely these dynamics by facilitating personalised learning paths, increasing academic independence, and providing accessible and adaptive educational support ([5]; [14]; [34]; [42]). Competence is strengthened when AI offers clear explanations and immediate, structured feedback; empirical evidence shows that high-quality AI output increases perceived usefulness, confidence, and academic self-efficacy ([5]; [18]; [25]; [34]; [42]). Finally, relatedness can be indirectly reinforced when AI reduces frustration and emotional overload, helping students engage more confidently with classroom or collaborative activities, especially as AI tools mitigate anxiety, sustain emotional engagement, and provide a sense of continuous support ([18]; [25]; [26]; [32]).

Critically, this supportive dynamic is not automatic. When AI outputs are unreliable, opaque, or difficult to interpret, they can frustrate these same psychological needs. This frustration can lead to anxiety, mistrust, or avoidance behaviours, patterns that align with the ambivalent student responses observed in empirical studies ([41]). Therefore, the motivational impact of AI in education is contingent; it hinges precisely on how the technology’s design and outputs either support or threaten the foundational needs for autonomy, competence, and relatedness.

### 2.3. Perceptions of Usefulness, Credibility, and Professional Relevance

The adoption and effective use of AI tools in education are not neutral acts, but are deeply influenced by students’ perceptions of this technology. The specialised literature argues that this perception is a critical determining factor for the meaningful integration of AI into learning processes ([10]). These perceptions, far from being homogeneous, are shaped by a systemic network of psychological, educational and contextual factors that interact with each other. Recent research highlights key dimensions such as perceived usefulness and credibility, professional relevance, prior technological capital, and psychosocial factors, including gender differences, as the main explanatory axes ([31]; [32]).

Among the most influential constructs is perceived usefulness, understood as the degree to which students believe that an AI tool improves their academic performance or efficiency. This perception is modulated by the credibility attributed to the system, that is, confidence in its accuracy and reliability, a factor that conditions attitudes and willingness to use these technologies ([5]; [18]). This relationship is particularly relevant in the context of generative language models such as ChatGPT, where the ability of students to critically evaluate the veracity and relevance of the output becomes a fundamental skill, given that perceived credibility conditions their attitudes towards the tool. Furthermore, previous experience with technology, such as familiarity with video games or programming, acts as an initial facilitator that predisposes students towards more positive attitudes ([18]).

In the specific field of engineering education, the factor of perceived professional relevance takes on particular importance. Students who see a clear and direct connection between mastering AI tools and the skills required in their future field of work show significantly greater motivation and willingness to integrate them into their learning ([24]; [31]). This perception transforms AI from just another academic tool into an integral component of their budding professional identity.

In addition to these cognitive and contextual factors, research shows that psychosocial variables such as gender play a significant modulating role. Studies in STEM disciplines consistently indicate that female students, even when exhibiting comparable or superior academic performance, tend to report lower technological self-efficacy, higher levels of anxiety when faced with unfamiliar digital tools, and a greater aversion to the risk of making public mistakes ([3]; [13]). These differences underscore the need for inclusive pedagogical approaches that address these perception gaps to ensure equitable adoption of AI.

In summary, engineering students’ perception of AI is a multifaceted and dynamic construct. It arises from the interconnection between the pragmatic assessment of its usefulness and credibility, the projection of its value in their professional careers, their prior knowledge and skills base, and the filter of individual psychosocial experiences. Understanding this systemic configuration, illustrated in Figure 1, is essential for designing educational interventions that not only introduce the technology but also cultivate the right perceptions for its responsible, critical, and effective use, thus preparing future engineers for a work environment increasingly mediated by AI.

## 3. Materials and Methods

### 3.1. Research Design

This study employed an exploratory mixed-methods design ([29]), in which quantitative and qualitative data were used in a complementary manner to obtain a broad understanding of students’ perceptions and experiences regarding the use of AI in higher education. Quantitative data were used to assess adoption levels, perceptions of usefulness, and gender-based differences, while the qualitative responses provided deeper insight into emotional tensions, perceived risks, and contextual challenges associated with AI use.

### 3.2. Participants and Sampling

The target population consisted of all first-, second-, and third-year students enrolled in the engineering programme at a Chilean university during 2024 (N ≈ 300). To ensure representativeness, a probabilistic stratified sampling strategy was used based on academic year. An optimal sample size of 170 participants was calculated using a 95% confidence level and a 5% margin of error. The final demographic composition of the sample (n = 170) was 57% male, 42% female, and 1% who preferred not to disclose their gender. Cohort distribution was balanced across academic levels, with 32% first-year students, 35% second-year, and 33% third-year students.

The decision to focus on this population is supported by recent evidence indicating that STEM students demonstrate higher predisposition to adopt emerging technologies such as AI, due to greater familiarity with computational environments and digital tools ([32]). Therefore, this population was considered particularly relevant to explore perceptions, experiences, and challenges associated with AI use in higher education.

Participation in the study was voluntary and anonymous, and all students provided informed consent electronically before accessing the questionnaire, in accordance with institutional ethical guidelines.

### 3.3. Instrument and Procedure

Data collection was carried out using a structured online questionnaire, created in Google Forms (Google LLC, Mountain View, CA, USA), with an estimated duration of 20 min. The instrument was designed ad hoc for this study and combined closed-ended questions (multiple choice and 7-point Likert scales) with open-ended prompts, enabling the simultaneous collection of quantitative and qualitative data. The questionnaire comprised three sections, each aligned with a specific research question:Patterns of use and adoption (RQ1): This section is primarily descriptive in nature. It aimed to quantify the penetration of AI tools among the student population, identifying the most popular platforms and exploring the primary reasons driving their use.Perception of usefulness and impact on learning (RQ2): After establishing the “what” and “why” of use, this section sought to assess the “how” its impact is perceived. Through Likert scales, students’ assessments of specific dimensions such as efficiency, autonomy, conceptual understanding, and the overall helpfulness of AI as an educational tool were measured. These items also allowed for the examination of potential gender-based differences in perceived usefulness.Overall opinion of AI (RQ3): This final section captured a more nuanced and comprehensive view of students’ experiences by asking them to describe the perceived benefits of using AI and the risks and disadvantages associated with its academic implementation.

Content validity was evaluated by three expert reviewers specialising in educational technology and psychometrics, who evaluated the clarity, relevance, and coherence of the questions. Minor adjustments were incorporated based on their recommendations. A pilot test with a small group of students (n = 16) verified item comprehension and approximate response time, resulting in no substantial modifications. Given the exploratory nature of the study, the instrument underwent expert review to ensure content validity; however, full psychometric validation (e.g., reliability analyses or construct validation procedures) was beyond the scope of the present research.

Finally, it is important to note that the questionnaire captured perceived engagement-related experiences rather than behavioural or performance-based indicators. Accordingly, all engagement-related results reflect students’ subjective evaluations of how AI supports or hinders their motivation, autonomy, and participation, rather than direct observations of engagement behaviours.

### 3.4. Data Analysis

Quantitative analyses were conducted in R software (version 4.3.3; R Foundation for Statistical Computing, Vienna, Austria) using descriptive and inferential statistics consistent with the nature of the variables and the objectives of the study.

For RQ1, univariate descriptive statistics (frequencies and percentages) were calculated to characterise patterns of AI tool usage among the student population.

For RQ2, a sequential analytical protocol was applied:Initial descriptive analysis: The means and standard deviations were calculated for each item on the Likert scale (Q1–Q10).Verification of parametric assumptions: Before comparing groups, the normality of the distribution of responses was evaluated using the Shapiro–Wilk test for each item, separately by gender. As none of the variables met the normality assumption (*p* < 0.05), non-parametric tests were used.Comparison of perceptions by gender: Statistically significant differences in perceptions between men and women were analysed using the Mann–Whitney U test for independent samples. This methodological choice was based on its independence from the assumption of normality.Frequency analysis and cross-tabulation of variables: A frequency analysis was performed to examine the relationship between self-reported technological familiarity and the perception that AI enhances autonomy in learning, by calculating frequencies and percentages.

The level of statistical significance was set at α = 0.05. For the first three analyses, calculations were performed considering only the categories “Male” and “Female,” excluding “Other” due to its small sample size, which prevented statistically robust comparisons.

To address RQ3, which examines the benefits, difficulties, risks, and emotions emerging from students’ experiences with AI, a thematic content analysis was conducted following [6]’s ([6]) guidelines, with support from NVivo 15 software (Lumivero, Denver, CO, USA).

Three researchers independently coded all open-ended responses and generated initial categories. These categories were subsequently compared, refined, and grouped into subthemes and overarching themes through constant comparison. Discrepancies were resolved through discussion until full consensus was achieved. Intercoder agreement, assessed at the level of initial categories, showed high conceptual coherence across coders.

Although the open-ended questions explicitly asked only about “benefits” and “difficulties/concerns,” their open formulation allowed the spontaneous emergence of additional dimensions relevant to RQ3, including emotional reactions (e.g., enthusiasm, curiosity, anxiety, frustration), perceived risks (e.g., plagiarism, reliability concerns), and tensions related to autonomy, dependence, and effort justification. These emergent dimensions were systematically incorporated into the coding scheme.

This inductive analytical strategy is consistent with exploratory mixed-methods designs, wherein categories derived from participants’ own language constitute a valid operationalisation of complex constructs not directly measured through closed-ended items. Thus, affective, ethical, and cognitive tensions, central to RQ3, were operationalised inductively through participants’ narratives rather than through predefined items, supporting a grounded understanding of ambivalence and emotional responses.

## 4. Results

The following section presents the results structured according to the three central dimensions of the questionnaire.

### 4.1. Use of AI Tools

73.5% of the students surveyed (n = 125) said they actively use AI tools to support their studies. Within this group, the gender distribution was balanced: 49.6% male, 49.6% female, and 0.8% who preferred not to specify.

Analysis by academic level showed a significant upward trend in AI adoption. The usage rate was 45% in the first year, increasing to 59% in the second year and reaching 72% in the third year. With this progression, continued exposure to the university environment and increased academic demands are linked to a greater willingness to adopt these technologies.

In terms of preferences, a clear predominance of natural language models was identified, with ChatGPT being the most popular platform, used by 80% of users. Substantially lower adoption rates were recorded for other tools: online tutoring platforms (e.g., Socratic) with 14.4%, knowledge engines for problem solving (e.g., Wolfram Alpha) with 11.2%, and assisted writing tools (e.g., QuillBot) with 1.6%. This pattern indicates that students favour general-purpose solutions over specialised tools.

Regarding the reasons for use, the three main reasons cited by students were: (1) to improve the quality of assignments and essays (38.4%), (2) to save time searching for information (36.8%), and (3) to facilitate the understanding of complex concepts (36.0%). These findings indicate that students value AI both for its pragmatic usefulness (time savings) and for its potential to improve the quality of their academic work.

### 4.2. Perception of AI Use and Learning

Students’ overall perception of the usefulness of AI in their learning was remarkably positive. As detailed in Table 1, items related to usefulness for understanding technical concepts and obtaining unique resources received the highest average scores (6.04 and 6.26, respectively). The lowest score was observed in the perception of content personalisation (4.69), suggesting a potential area for improvement for this type of tool.

The comparison of perceptions by gender (Table 2) showed that, of the ten dimensions evaluated, only three presented statistically significant differences (*p* < 0.05). Female students rated the overall usefulness of AI for their learning (Q1), its help in understanding technical concepts (Q4), and access to unique resources (Q8) significantly higher than males. In the remaining seven dimensions, the differences did not reach statistical significance, indicating a largely homogeneous perception between genders.

Finally, when analysing the relationship between self-reported technological familiarity and the perception that AI enhances autonomy in learning, a direct and marked association was observed. While 83% of students with high technological familiarity reported an increase in their autonomy, this percentage decreased to 67% among those with a medium level of familiarity and further to 40% among those with low familiarity. These results suggest that the perception of autonomy associated with the use of AI depends mainly on the student’s level of confidence and technological competence, reinforcing the idea that the effective use of these tools is linked not only to their availability but also to the digital skills that mediate their use.

### 4.3. General Opinion Regarding AI

Analysis of the open-ended responses identified six key benefits that students associate with the use of AI:Efficiency and time savings: Students consistently emphasised AI’s ability to optimise study time by providing rapid solutions to conceptual or procedural questions. Several indicated that it “allows me to move forward more quickly” and highlighted its “efficiency and speed”. Others noted that AI helps them “save time” and concentrate on core learning tasks instead of spending long periods searching for information.24/7 availability and access: The possibility of having a perpetual learning resource was highly valued. Many students described AI as an “always-available tutor,” emphasising its “24/7 availability to resolve doubts”. This constant access enabled them to study whenever difficulties arose, without depending on teacher schedules, and to “learn whenever I want.”Support for learning and content comprehension: A recurring benefit was AI’s capacity to clarify and explain difficult content. Students reported that it helps them “understand difficult concepts in a different way” and “follow step-by-step processes to solve problems.” They also appreciated the availability of alternative explanations that complement classroom learning, noting that AI “gives me examples that make sense to me.”Improvement of their own work: Students recognised AI as a tool for improving the quality of their academic output. Several highlighted that it supports them in “organising my thinking when I don’t know where to start,” refining their writing, or verifying calculations. These functions help them produce clearer and more coherent work while facilitating the initiation of challenging tasks or projects.Strengthening autonomy: Responses indicated that AI fosters a sense of empowerment and self-directed learning. Students described that the tool enables them to “research on my own,” make informed decisions, and manage their learning more effectively. For many, this autonomy translates into taking greater responsibility for their academic progress.Exploration and expansion of knowledge: Some participants valued the opportunity to explore content beyond the formal curriculum. AI was perceived as a catalyst for intellectual curiosity, allowing them to delve into topics of personal interest and to “broaden my academic horizons” through guided exploration.

The identified benefits show that students perceive AI as a multidimensional facilitator that optimises not only their operational efficiency but also their capacity for understanding, autonomy, and intellectual exploration, transcending its merely instrumental function.

Furthermore, five main risks or disadvantages also emerged:Dependence and weakening of cognitive skills: One of the most recurrent concerns was the possibility of becoming dependent on the tool, potentially leading to reduced autonomy and the weakening of skills such as critical thinking, deep reflection, and the construction of one’s own academic voice. Several students worried that AI might “think for them,” while others expressed that they “feel insecure if I don’t use it.” These concerns illustrate a perceived risk that AI could replace rather than support authentic cognitive effort.Risk of plagiarism and ethical dilemma: Students demonstrated strong awareness of ethical risks. Concerns about plagiarism and the feeling of “cheating” were prominent, with some expressing fear of inadvertently misusing the tool or violating academic rules. This reflects a tension between the desire to leverage a powerful resource and the responsibility to maintain academic integrity. As one student noted, “I worry about accidentally doing something that could be considered plagiarism.”Reliability of information: A frequently mentioned disadvantage involved the accuracy and depth of AI-generated information. Students reported varying levels of distrust and described having to invest additional time verifying outputs, which often generated frustration. Several expressed “the anxiety of not knowing whether the information is reliable” or feeling unsettled because AI “might be making things up.” This paradox, saving time but losing time in fact-checking, appeared repeatedly in their responses.Barriers to use and technical complexity: Some students pointed to practical difficulties in interacting effectively with AI systems, particularly regarding the formulation of queries or the initial learning curve. The frustration caused by unclear prompts or unexpected results was reflected in comments such as “the frustration when it doesn’t give me the result I expect” and the sense that “it is complex, and I don’t have the habit yet.” These responses highlight the cognitive and technical demands of effective use.Contextual and access limitations: Students also noted inequalities in access and digital preparedness. Not all participants felt they possessed the same level of digital skills or familiarity with emerging technologies. Some perceived AI as too specialised for certain subjects or insufficient to replace traditional resources such as textbooks, group work, or teacher explanations. As one student shared, “Sometimes I prefer books or classmates; AI doesn’t always work for every topic.” Another emphasised the epistemic limits of AI by noting that “AI is not a primary source,” highlighting its inability to substitute foundational academic materials.

These disadvantages reveal that the implementation of AI in academia presents significant challenges that go beyond the technical, encompassing ethical, cognitive, and social dimensions. The paradox between efficiency and verification, along with concerns about cognitive dependence, points to the need to develop guidelines for critical and responsible use.

Analysing perceptions from a gender perspective, a significant asymmetry was found in how students value AI in terms of its usefulness versus complexity. Forty-nine percent of men and 65% of women said they consider AI to be a “useful but difficult to apply” tool. This divergence suggests that there are gender-differentiated factors in the perception of AI’s ease of use, which could indicate the need to address gaps in trust or digital competence specifically.

Despite these risks, the affective component of interaction with AI, summarised in Figure 2, shows a predominance of positive emotions. The most frequent were enthusiasm, confidence, curiosity, and satisfaction. However, a significant portion of the sample also reported negative emotions, such as anxiety and frustration. The coexistence of positive and negative emotions reflects the ambivalent nature of the AI experience, where enthusiasm for its potential coexists with anxiety generated by the practical and ethical challenges it presents.

Taken together, these perceptions reveal that students’ use of AI does not depend solely on technological access, but is also conditioned by the availability of training support and institutional guidance. Concerns about plagiarism, cognitive dependency, information verification, and the perception of technical skill gaps among students, particularly associated with gender differences and levels of technological familiarity, suggest that there is an explicit need for structured support and the development of critical skills for the responsible and autonomous use of these tools. These findings provide a basis for arguing that the effective and ethical adoption of AI in educational contexts requires institutional conditions that transcend its technical availability and connect with training needs expressed directly by the students themselves.

## 5. Discussion

The results of this study paint a nuanced and multidimensional picture of the adoption, perceptions, and challenges associated with the use of AI tools among engineering students at a Chilean university. The findings not only corroborate trends reported in the international literature but also reveal particularities specific to the Latin American context that require particular attention.

A notable initial finding is the high level of AI adoption (73.5%), confirming its penetration in higher education and the predisposition of STEM areas towards emerging technologies ([32]). This phenomenon is nuanced by a clearly upward adoption pattern according to academic year, increasing from 45% in the first year to 72% in the third. This progression suggests that adoption does not depend solely on temporary exposure, but on the parallel development of academic maturity and self-regulated learning skills ([39]). As students progress and face more complex demands, they become more capable of strategically managing their learning resources, including AI. This development aligns with [2]’s ([2]) theory of self-efficacy, as progressive mastery of the university environment strengthens students’ confidence, enabling them to integrate AI more effectively and critically.

Regarding usage preferences, the overwhelming preference for general-purpose tools such as ChatGPT over specialised platforms indicates that students primarily value versatility and accessibility over specific functions tailored to disciplines. This observation coincides with the findings of [18] ([18]), who highlight that perceived usability and ease of access are key determinants in the early adoption of educational technologies.

In terms of perceived usefulness, students recognise the value of AI for understanding technical concepts and accessing unique resources, reinforcing its role as a facilitator of deep learning. Significantly, while no significant gender differences were observed in most dimensions, female students reported significantly more favourable perceptions than their male peers in three key areas: overall usefulness (Q1), understanding technical concepts (Q4), and access to unique resources (Q8). These findings suggest that women may place significantly higher value on the potential of AI as a learning support resource, possibly due to a heightened search for complementary strategies to strengthen their understanding in a traditionally male-dominated field such as engineering. However, the low score on the personalisation dimension (4.69) reveals a substantial limitation of current tools, which still fail to fully adapt to individual learning needs, as noted by [14] ([14]). This gap between expectation and reality may be a factor that limits the pedagogical potential of AI in the absence of more refined and contextualised designs.

Qualitative analysis enriches and complicates this narrative, revealing a fundamental duality in student perception. On the one hand, AI is conceptualised as an “artificial companion” that facilitates behavioural and emotional engagement in autonomous study, ensuring continuity when no human guidance is available. This assessment coincides with studies that highlight its ability to offer immediate help in resource-constrained environments ([21]) and to improve student engagement when integrated as a continuous support resource ([42]).

On the other hand, students clearly articulate the counterweight that sustains their ambivalence: a marked ethical and functional tension. The conceptualisation of AI as a “quick fix” highlights the risk of instrumentalisation, which can erode cognitive autonomy and encourage superficial engagement in the absence of critical mediation. In this vein, [19] ([19]) warns of the need to maintain a balance between interpersonal relationships and technological interventions: although AI-driven solutions can offer highly individualised learning experiences, they should be used as a complement to, rather than a substitute for, human intervention, given that human involvement remains essential for fostering critical thinking, creativity and socio-emotional skills.

The tension is also evident in the mistrust and anxiety expressed by students, who question the reliability of the answers, fear “copying without thinking” and experience a persistent feeling of “cheating”. These concerns coincide with those raised by [41] ([41]), who warn that generative AI systems cannot be considered fully reliable sources of information, as they often produce content that is seemingly coherent but potentially incorrect. This well-founded scepticism, coupled with the lack of clear regulations which, according to [28] ([28]), generates confusion and fear of being judged, acts as a significant barrier to authentic and sustained engagement. Taken together, the perception of intellectual and ethical risk (due to accusations of plagiarism and an ambiguous regulatory framework) not only reduces the frequency and intention of use, as these authors point out, but also encourages an evasive or superficial relationship with the tools, thus preventing the deep and critical immersion that characterises truly transformative learning.

A more integrated interpretation emerges when drawing on SDT. SDT helps explain why ambivalence exerts such a strong influence on engagement: students’ positive experiences with AI, such as independence, efficiency, improved comprehension, and 24/7 availability, signal potential satisfaction of autonomy and competence needs. However, the concerns voiced by participants, such as anxiety about plagiarism, distrust of the information generated, and cognitive dependence, reflect frustration of these same needs. This duality is consistent with the emotional ambivalence described in the theorical framework and aligns with broader research on AI-mediated learning dynamics. In this sense, SDT offers a coherent interpretive lens for understanding how the same technology can simultaneously empower and constrain students, depending on how it interacts with their psychological needs.

From this perspective, a crucial question arises: to what extent are increases in autonomy or engagement attributable to AI itself, and to what extent do they reflect pedagogical conditions that, regardless of technology, also enable the satisfaction of autonomy and competence needs? AI may amplify processes such as scaffolding, explanation, and feedback, but it does not inherently produce motivational benefits that cannot be achieved through other instructional designs. Extensive evidence demonstrates that active learning, structured feedback, and collaborative pedagogies can likewise enhance autonomy, competence, and motivation ([7]; [35]; [40]). Thus, the present findings should be interpreted not as evidence of AI’s unique motivational power but as indicators of how it can operate as a facilitator within a broader pedagogical ecosystem.

Building on this interpretation, the findings point to a broader institutional responsibility. If students’ engagement with AI is shaped by the dynamic interplay between ambivalence, perceived usefulness, and the satisfaction or frustration of autonomy and competence needs then institutional strategies must address not only access to technology but also the emotional, cognitive, and equity-related conditions that shape these experiences. In this sense, promoting critical digital literacy becomes essential. Such literacy should be conceived not merely as technical training, but as a holistic educational process that integrates sound instructional design, explicit practices of verification and evaluation of AI-generated content, and a strong ethical grounding ([25]; [26]). This approach aligns with international principles for the responsible use of AI, such as those promoted by [37] ([37]) and the [27] ([27]), which emphasise transparency, equity, human oversight, and accountability.

## 6. Conclusions

This study examined the adoption, perceptions, and challenges associated with the use of AI tools among engineering students in Chile, offering a nuanced view of how generative systems are reshaping learning experiences in higher education. The findings reveal high levels of adoption and strong recognition of AI’s academic value, particularly as a resource for understanding complex concepts, supporting autonomous learning, and providing continuous access to personalised academic assistance. At the same time, this enthusiasm coexists with mistrust, anxiety, uncertainty about accuracy, and concerns related to academic integrity, illustrating the ambivalence that characterises students’ engagement with AI.

Three key implications emerge from these results. First, the development of critical digital literacy is essential to ensure that students use AI not merely as a shortcut for task completion, but as a catalyst for analytical reasoning, reflective learning, and responsible decision-making. Second, institutions need clear and transparent ethical guidelines that reduce fears of punitive consequences, especially regarding plagiarism, and foster a safe environment for the responsible and pedagogically grounded use of AI.

Third, and most importantly, the findings highlight the relevance of a multidimensional institutional support strategy. Beyond providing access to technological tools, universities must address the emotional, cognitive, and social dimensions identified in the qualitative evidence. This includes pedagogical accompaniment that reduces anxiety and uncertainty, training that mitigates competence gaps, including gender-related disparities, and structured opportunities for collaborative learning that promote trust and shared skill development. These recommendations stem directly from the tensions expressed by students, who articulated appreciation for AI as a learning companion alongside fears of dependency, inaccuracy, and uneven familiarity with prompting techniques.

Practically, the results underscore the need to design AI tools and pedagogical practices that support differentiated learning needs and reduce confidence gaps. Incorporating adaptive scaffolding, feedback-explanation features, and reflective activities can promote more equitable engagement and ensure that the benefits of AI are accessible to students regardless of gender or technological familiarity.

A central contribution of this study lies in showing that the pedagogical value of AI does not depend on the frequency of use alone but on the extent to which learning environments support students’ psychological needs related to autonomy, competence, and belonging. The coexistence of enthusiasm and anxiety, together with the perceived gender differences in confidence, suggests the emergence of an “AI engagement gap,” an uneven distribution of benefits shaped by emotional comfort, prompting skills, and institutional mediation. This insight contributes to broader debates in engineering education and technology-enhanced learning, where equitable and responsible adoption of emerging technologies remains a central challenge.

In the Latin American context, marked by structural inequalities and variable access to digital resources, recognising and addressing this engagement gap is particularly relevant. The findings presented here reinforce the idea that AI can democratise learning or intensify inequality depending on how it is implemented, mediated, and supported. By situating students’ experiences at the centre of AI integration, this study offers evidence that can inform policies, curriculum design, and pedagogical strategies that promote ethical, inclusive, and student-centred innovation in higher education.

## 7. Limitations and Directions for Future Research

Although the results offer relevant insights, this study has several limitations that should be acknowledged. First, the sample comes from a single institution and focuses exclusively on engineering programmes, which limits the possibility of generalising the findings to other disciplinary, institutional or cultural contexts.

Secondly, although gender differences were examined, this analysis is exploratory in nature and was conditioned by sample imbalance, which prevented more robust comparisons. Consequently, the gender-related patterns identified in this study should be interpreted as indicative trends rather than generalisable conclusions.

Thirdly, the cross-sectional design and reliance on self-report measures may introduce various biases, such as social desirability bias and over- or underestimation of certain behaviours. Some students may have emphasised the positive uses of AI or downplayed problematic practices to align with academic norms perceived as socially acceptable.

Furthermore, the questionnaire captured perceived experiences of participation rather than behavioural or observational indicators. Therefore, the results related to engagement reflect only subjective assessments of how AI supports or hinders student motivation, autonomy, and participation, without directly measuring actual learning behaviours or academic performance. This is an inherent limitation of studies based exclusively on self-reported perceptions.

Fourthly, although qualitative responses provided valuable information about emotional tensions, ethical perceptions, and dilemmas associated with the use of AI, their nature, based on open self-reports, limits interpretative depth. Additional qualitative methods, such as semi-structured interviews or focus groups, could broaden our understanding of these experiences, capturing more complex nuances and allowing us to explore how these dynamics manifest themselves in different educational contexts.

Finally, future research could evaluate pedagogical or institutional interventions aimed at promoting ethical, equitable, and pedagogically meaningful integration of AI. This includes critical digital literacy programmes, support strategies that address gender-related trust gaps, or design experiments that test adaptive scaffolding features in AI tools. Likewise, longitudinal or experimental studies comparing courses that integrate AI with courses that employ traditional active learning approaches would help determine whether the benefits observed in student engagement are due to specific AI functionalities or broader pedagogical conditions.

## Figures and Tables

**Figure 1 behavsci-16-00011-f001:**
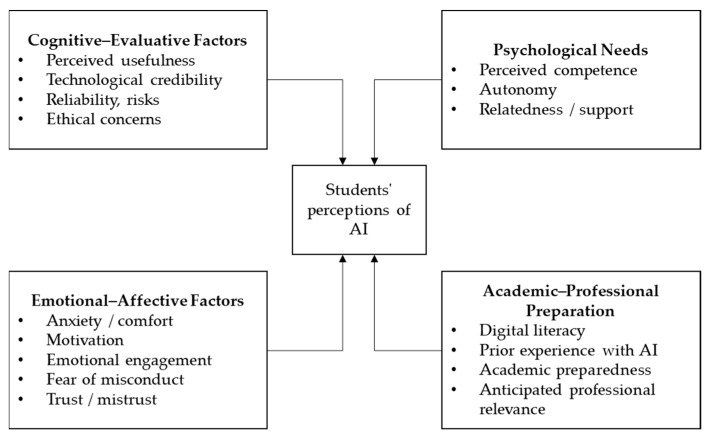
Factors influencing students’ perceptions.

**Figure 2 behavsci-16-00011-f002:**
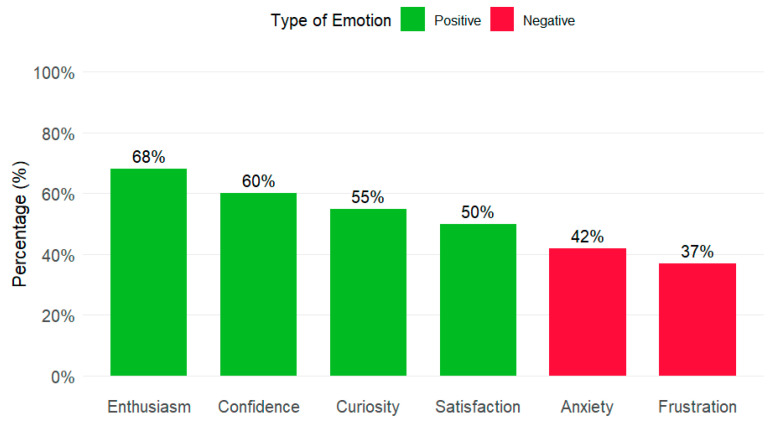
Associated emotions with the use of AI (%).

**Table 1 behavsci-16-00011-t001:** General perceptions about AI and learning.

Code	Item	Mean	SD
Q1	AI has been helpful for my learning.	6.05	0.80
Q2	Learning with AI is more efficient than traditional learning methods.	5.52	1.03
Q3	I believe that incorporating AI into my learning provides me with a distinct advantage over traditional study methods.	5.67	0.87
Q4	AI helps me better understand the technical concepts of my major.	6.04	0.80
Q5	AI allows me to solve problems more efficiently than other methods.	5.63	1.00
Q6	AI helps me stay up to date on the topics I study.	5.48	0.99
Q7	Using AI enables me to learn more dynamically and interactively.	5.69	0.84
Q8	AI provides me with learning resources that I would not find with traditional methods.	6.26	0.62
Q9	AI tools personalise educational content according to my learning needs.	4.69	0.84
Q10	AI encourages my independence in learning, without the need for constant support from teachers.	5.67	0.87

**Table 2 behavsci-16-00011-t002:** Perceptions of AI and learning by gender.

Code	Male		Female		W	*p*-Value
	Mean	SD	Mean	SD		
Q1	5.77	0.80	6.32	0.72	1217	0.00
Q2	5.52	1.07	5.53	1.00	1933	0.96
Q3	5.63	0.91	5.73	0.83	1811	0.55
Q4	5.77	0.80	6.31	0.71	1237	0.00
Q5	5.58	1.11	5.68	0.90	1866.5	0.77
Q6	5.42	1.06	5.55	0.92	1819.5	0.59
Q7	5.63	0.91	5.76	0.76	1796	0.50
Q8	6.06	0.64	6.45	0.54	794.5	0.00
Q9	4.63	0.91	4.76	0.76	1796	0.50
Q10	5.63	0.91	5.73	0.83	1811	0.55

## Data Availability

The original contributions presented in the study are included in the article; further inquiries can be directed to the corresponding author.

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
