# Peer review of "Navigating Ambivalence: Artificial Intelligence and Its Impact on Student Engagement in Engineering Education"

_behavsci, 2025, doi:10.3390/bs16010011_

Round 1

Reviewer 1 Report

Comments and Suggestions for Authors

Please see attachment. Interesting work!

Author Response

Comment 1: Findings: Clearly presented although looking at the survey questions it is difficult to understand why institutional strategy is brought to the forefront in the conclusions, since this is not really asked about. The operational aspects concerning learning is assessed, yet it remains unclear how critical digital literacy, clear policies, and the support for programmes connecting to the differences in skill prerequisites and gender gaps are influencing this outcome. Although those aspects are important the data is presenting perceptions about AI and learning with also gender differences, the implications are therefore weak, or even more of a post analysis construction, which to the reviewer is less connecting to the data, and those more speculative.

Response 1: We sincerely thank the reviewer for this constructive observation. In response, we revised the manuscript to explicitly articulate how the implications discussed stem from the empirical findings rather than speculative reasoning. Specifically, in Section 4.3 (Results), we added a new paragraph that directly links students’ perceptions of risks, anxiety, confidence asymmetries, and gender differences with the need for institutional actions such as critical digital literacy training and differentiated support. Furthermore, in the Discussion and Conclusion sections, we reformulated the institutional implications to ensure that they are clearly grounded in the qualitative evidence, emphasising that the recommended multidimensional support strategy derives from students’ reported experiences of ambivalence and uneven competence rather than external assumptions. These revisions strengthen the internal coherence between findings and implications and avoid any perception of post-hoc interpretation.

Comment 2: Discussion: By drawing attention to anxiety and tensions, plus gender differences there could have been supported by more in-depth discussions analysing these aspects in light of SDT more thoroughly. In this version the connection is mainly based on the fact that students are transformed to (row 354) : “active managers of their learning”.

Although the reviewer does not disagree with this, as for a contribution to the scholarly knowledge wouldn’t it be fair to reason or at least considering a what a reference group would could have contributed in terms of perspectives. Saying that AI, and the engagement boost it offers, or could it be that this could also happened if a pedagogy and learning approaches could be equally important?

Response 2: Thank you for this valuable suggestion. The Discussion has been substantially expanded to deepen the application of Self-Determination Theory (Deci & Ryan, 2000; Ryan & Deci, 2017) and to analyse anxiety, tensions, and gender-related differences through the lenses of autonomy, competence, and relatedness. In addition, we incorporated a new analytic section questioning whether the reported engagement benefits can be attributed uniquely to AI or whether similar outcomes might occur through high-impact pedagogical approaches without AI. This addition includes references to comparative pedagogical research and proposes future studies with reference groups to evaluate AI’s distinctive contribution. These revisions strengthen theoretical integration and enhance the scholarly contribution of the study.

Comment 3: Conclusion: This section is well aligned with the purpose, presenting how Chilean students are adopting and practicing AI to improve their academic work. The three important findings for institutional adoption involve three pillars, where the third, a multidimensional strategy to successfully reach the objective for learning activities remains somewhat unclear and loosely motivated.

This also impact the somewhat limited novel insights that the study adds to the ongoing debate that given the contextual contribution, still could potentially enrich more than mere “whose real pedagogical value depends on the student's ability to use it critically, reflectively and in a self regulated manner.” (row 439 440).

Response 3: We appreciate the reviewer’s insightful feedback. The Conclusion section has been rewritten to clarify and substantively develop the third pillar (the multidimensional institutional support strategy). It now explicitly explains its components—pedagogical accompaniment, differentiated training to mitigate competence and gender-related gaps, and collaborative learning structures—and directly connects them to evidence from qualitative results. Additionally, to enhance the contribution to the ongoing debate, a new conceptual insight, the “AI engagement gap,” has been introduced to capture the uneven benefits students experience depending on confidence, emotional comfort, and institutional mediation. This reframing strengthens the originality and relevance of the conclusions.

Comment 4: Minor typos:

Row 262

4.3 General opinion regarding IA

Response 4: Thank you for noting this. The subtitle has been corrected.

Row 484

References

Systematically using citation with full text reference, this is not correct, please update by taking out ex. (Al-Abdullatif & Alsubaie, 2024) etc.

Response 5: We appreciate the reviewer’s careful attention to formatting. The reference list has been corrected to remove in-text citation formatting (e.g. parentheses), ensuring consistency with journal guidelines.

(THE MODIFICATIONS MADE TO THE ORIGINAL MANUSCRIPT, FOLLOWING THE REVIEWERS' OBSERVATIONS, ARE HIGHLIGHTED IN RED)

Reviewer 2 Report

Comments and Suggestions for Authors

Thank you for submitting your manuscript, “Navigating Ambivalence: Artificial Intelligence and its Impact on Student Engagement in Engineering Education.” The topic is highly relevant and addresses an important emerging issue in engineering education. Your work offers valuable insights into how student ambivalence shapes engagement with AI-supported learning environments.

Overall, the manuscript shows promise, but several areas require further refinement to enhance clarity, coherence, and methodological rigour. In particular, improvements are needed in the clarity of the abstract, structure of the introduction, integration of the theoretical framework, detail in the methods, interpretation of results, and consistency of language.

A comprehensive section-by-section evaluation—covering all the sections—is provided in the attached detailed review report. Please review that document carefully, as it contains specific guidance and recommendations to support your revision.

In summary, the manuscript has strong potential but requires minor revisions to strengthen its theoretical grounding, methodological transparency, and overall academic presentation.

Comments on the Quality of English Language

Meticulous proofreading is required 

Author Response

Comment 1: Abstract: The abstract provides a concise overview of the study with some appropriate structure. It highlights the central theme of ambivalence toward AI and outlines the methodological approach.

Suggestions for Improvement: It would benefit from clearer articulation of the research gap and a more explicit statement of key findings and main conclusions would strengthen clarity and impact.

Response 1: We thank the reviewer for this constructive suggestion. In response, we revised the Abstract to explicitly articulate the research gap concerning the limited empirical evidence on students’ emotional, cognitive, and ethical experiences with AI, particularly in Latin American contexts. We also added a clearer summary of the key findings, including the coexistence of enthusiasm and anxiety, gender-related differences in confidence, and the identification of an emerging “AI engagement gap.” Finally, we strengthened the closing statement to highlight the main conclusions regarding the need for critical digital literacy, ethical guidelines, and multidimensional institutional support for equitable AI adoption.

Comment 2: Keywords: Relevant but Required reconsideration to include one or two more.

Suggestions for Improvement: You might consider adding, for example, technology acceptance, learning analytics, or digital pedagogy to improve indexing and discoverability. Keyword students is generic, combine Artificial Intelligence with abbreviation, seems two different.

Response 2: Thank you for this helpful suggestion. In response, we revised the Keywords to enhance specificity, indexing, and alignment with the study’s focus. We incorporated terms such as technology acceptance and digital pedagogy, and combined “Artificial Intelligence” with its abbreviation “AI” to improve clarity, following the reviewer’s recommendation. We also replaced the generic keyword “students” with more specific descriptors related to engagement and gender differences.

Comment 3: Introduction: Strengths: The introduction effectively establishes the rationale and context for the study. It addresses the growing influence of AI in higher education and appropriately frames the concept of student ambivalence. It situates the study within contemporary debates around technology-enabled learning.

Suggestions for Improvement: The argument would be stronger with a clearer articulation of the research problem/Gap and explicit research questions. Some paragraphs could be tightened to reduce repetition. Please also provide justification/rationale for the study.

Response 3: Thank you for the insightful suggestion. We reviewed and rewrote the Introduction to clearly articulate the research gap and rationale, added explicit research questions aligned with the study’s purpose, and tightened the narrative to reduce repetition. These modifications strengthen the theoretical coherence and clarity of the manuscript.

Comment 4: Theoretical Framework: Well-organized structure, it draws on relevant theoretical perspectives (e.g., technology acceptance, cognitive engagement, ambivalence theory). The discussion is conceptually grounded, but integration across theories could be improved.

Suggestions for Improvement: Some constructs appear underexplained or disconnected from later analyses. A clearer conceptual model (graphical representation) would help readers understand how variables and relationships were operationalised.

Response 4: Following the reviewer’s recommendation, we developed a conceptual model (Figure 1) that visually represents how these four dimensions (cognitive-evaluative factors, emotional–affective dynamics, psychological needs, academic and professional preparation) interact to shape students’ perceptions of AI. This graphical representation enhances conceptual transparency and facilitates readers’ understanding. We appreciate the reviewer’s comment, which significantly contributed to improving the coherence and clarity of the manuscript.

Comment 5: Materials and Methods: Methods are generally clear and systematic. The methodological approach is generally appropriate for the research aims. The description of participants, sampling strategy, and instruments is adequate,

Suggestions for Improvement: Additional detail regarding measurement validity and reliability would enhance methodological rigour. The procedure section should clarify how ambivalence was measured and whether potential confounders were controlled. Ethical considerations should be more explicitly stated.

Response 5: Thank you for this helpful observation. We have strengthened the Materials and Methods section to enhance methodological clarity and rigour. First, we expanded the description of the validation process of the instrument, including expert review and pilot testing. Second, ethical considerations were made more explicit by specifying the voluntary and anonymous nature of participation and the provision of informed consent prior to accessing the questionnaire.

Regarding the control of potential confounders, this study did not employ a confounder-control design, as it was not part of the methodological framework. However, the strengthened descriptions of sampling, measurement, and analytical procedures provide a transparent and coherent account of the study’s approach. Together, these revisions address the reviewer’s concerns and improve the methodological clarity of the manuscript.

Comment 6: Results: The results are presented clearly with appropriate tables and figures. The statistical analyses appear suitable.

Suggestions for Improvement: No improvements are required

Response 6: Thank you very much for your comment.

Comment 7: Discussion: The discussion effectively situates the findings within existing literature and highlights the dualistic nature of student perceptions of AI. The practical implications for engineering educators and curriculum designers are meaningful

Suggestions for Improvement: Some claims require stronger evidence or citation support. The manuscript would benefit from deeper analysis of cultural, disciplinary, or contextual factors influencing ambivalence.

Response 7: Thank you for this valuable observation. In the revised manuscript, the Discussion section has been substantially strengthened. We expanded the integration of supporting evidence and added further citations to reinforce the claims made.

Comment 8: Conclusion: The conclusion summarizes the main findings of the study.

Suggestions for Improvement: However, the section could be sharper by explicitly linking conclusions to broader debates in engineering education and AI adoption. Future research suggestions are appropriate but could be expanded to include longitudinal or comparative studies.

Response 8: We appreciate the reviewer’s thoughtful observation. In the revised manuscript, the Conclusion section has been strengthened by explicitly linking the findings to broader debates in engineering education and AI adoption, highlighting issues such as autonomy, competence, equity, and the emerging “AI engagement gap.” Furthermore, the Limitations and Directions for Future Research section now includes explicit recommendations for longitudinal, comparative, and intervention-based studies, directly addressing the reviewer’s suggestion.

Comment 9: Additional Suggestions: The manuscript is generally well written and clearly structured. A few grammatical inconsistencies, tense shifts, and minor phrasing issues remain. A thorough proofreading is recommended before final submission.

Response 9: Thank you for this helpful comment. In response, the entire manuscript has been carefully reviewed and professionally proofread by a specialist in academic English. Grammatical inconsistencies, tense shifts, and phrasing issues identified in the prior version have been corrected to improve clarity, readability, and stylistic consistency throughout the text. We appreciate the reviewer’s observation, which contributed to enhancing the overall quality of the manuscript.

Comment 10: Minor Revision: This manuscript addresses a timely and relevant issue in engineering education: the complex interplay between AI adoption and student engagement. The study offers meaningful insights but would benefit from refinements in theoretical integration, methodological detail, and clarity of results interpretation.

Response 10: We sincerely thank the reviewer for this encouraging and constructive observation. In response, the manuscript has been thoroughly reviewed to strengthen the areas highlighted. These revisions enhance the overall precision, transparency, and conceptual clarity of the manuscript.

(THE MODIFICATIONS MADE TO THE ORIGINAL MANUSCRIPT, FOLLOWING THE REVIEWERS' OBSERVATIONS, ARE HIGHLIGHTED IN RED)

Reviewer 3 Report

Comments and Suggestions for Authors
  1. Clarify Methodological Details: While the mixed-methods approach and triangulation are commendable, providing more explicit details about the data collection procedures, such as sampling strategy and the validation process for the questionnaire, would enhance transparency and reproducibility.

  2. Address Potential Biases: Consider discussing possible biases, such as self-report bias or social desirability effects, that could influence participants’ responses, especially in self-assessed perceptions of AI usefulness and ethical concerns.

  3. Deepen the Ethical Discussion: The manuscript highlights ethical ambivalence toward AI; expanding this discussion to include specific frameworks or guidelines for ethical AI use in educational contexts could strengthen the practical implications.

  4. Expand on Practical Recommendations: Based on findings about gender differences and perceptions, providing more concrete recommendations for designing AI tools that address current limitations (like personalization) and foster critical engagement would be valuable for practitioners.

  5. Language and Clarity: Some sections contain dense or complex sentences; simplifying language and enhancing clarity would improve accessibility for a broader audience.

  6. Figures and Tables: Ensure that all tables and figures are fully self-explanatory, with clear titles, labels, and legends, to facilitate quick comprehension of the results.

  7. Future Directions: Highlight potential avenues for future research, such as longitudinal studies or interventions to promote ethical AI use, to inspire ongoing scholarly contributions.

Author Response

Comment 1: Clarify Methodological Details: While the mixed-methods approach and triangulation are commendable, providing more explicit details about the data collection procedures, such as sampling strategy and the validation process for the questionnaire, would enhance transparency and reproducibility.

Response 1: We thank the reviewer for this valuable suggestion. In response, we expanded the Methodology section to provide detailed information on the sampling strategy, data collection procedures, and the validation process of the questionnaire. These additions improve methodological transparency and enhance the reproducibility of the study.

Comment 2: Address Potential Biases: Consider discussing possible biases, such as self-report bias or social desirability effects, that could influence participants’ responses, especially in self-assessed perceptions of AI usefulness and ethical concerns.

Response 2: Thank you for this valuable comment. We have now explicitly addressed potential sources of bias, including self-report bias and social desirability effects, which may have influenced participants’ responses regarding AI usefulness and ethical considerations. This discussion has been integrated into the Limitations and Directions for Future Research section to strengthen methodological transparency and clarify boundaries for interpretation.

Comment 3: Deepen the Ethical Discussion: The manuscript highlights ethical ambivalence toward AI; expanding this discussion to include specific frameworks or guidelines for ethical AI use in educational contexts could strengthen the practical implications.

Response 3: We appreciate this valuable recommendation. In response, we have expanded the ethical discussion by incorporating internationally recognised frameworks and guidelines for responsible AI use in education (UNESCO, 2021; OECD, 2019). A new paragraph has been added to the Discussion section to connect students’ ethical ambivalence to these principles and to outline practical implications for institutional decision-making and pedagogical strategies. This strengthens the conceptual depth and applied relevance of the manuscript.

Comment 4: Expand on Practical Recommendations: Based on findings about gender differences and perceptions, providing more concrete recommendations for designing AI tools that address current limitations (like personalization) and foster critical engagement would be valuable for practitioners.

Response 4: We thank the reviewer for this thoughtful suggestion. In response, we have expanded the Conclusions to include specific and actionable recommendations for the design and implementation of AI tools that address gender-related confidence gaps, support personalised scaffolding, and foster critical engagement. These additions strengthen the applied contribution of the study and provide practical implications for educators and system designers.

Comment 5: Language and Clarity: Some sections contain dense or complex sentences; simplifying language and enhancing clarity would improve accessibility for a broader audience.

Response 5: We appreciate this feedback. In response, we have thoroughly revised the manuscript to simplify the language and break down complex sentences, thereby improving clarity and accessibility for the reader.

Comment 6: Figures and Tables: Ensure that all tables and figures are fully self-explanatory, with clear titles, labels, and legends, to facilitate quick comprehension of the results.

Response 6: Thank you for this observation. We have reviewed all figures and tables to ensure they are fully self-explanatory.

Comment 7: Future Directions: Highlight potential avenues for future research, such as longitudinal studies or interventions to promote ethical AI use, to inspire ongoing scholarly contributions.

Response 7: Thank you for this insightful suggestion. In response, we expanded the Limitations and Directions for Future Research section to explicitly outline potential avenues for future research, including longitudinal and intervention-based studies, comparative designs examining pedagogical strategies with and without AI, and applied research focused on ethical and equitable AI integration. These additions strengthen the forward-looking contribution of the manuscript and align closely with the reviewer’s recommendation.

(THE MODIFICATIONS MADE TO THE ORIGINAL MANUSCRIPT, FOLLOWING THE REVIEWERS' OBSERVATIONS, ARE HIGHLIGHTED IN RED)

Reviewer 4 Report

Comments and Suggestions for Authors

This study aims to analyze the perceptions of AI tool usage and its impact on student engagement among 170 engineering students at a university in Chile, using a mixed-methods approach. Considering the recent surge of research on the use and effectiveness of AI in education, and the timeliness of the topic, the research focus and purpose may hold a degree of scholarly relevance. However, substantial discussion and improvement are required regarding the following issues.

  1. Although the study claims educational effects of AI usage experience, it lacks quantitative indicators that directly measure learning engagement or academic outcomes. While the concept of engagement is emphasized, the analysis relies solely on perception-based survey items, without incorporating behavioral or performance-based data (e.g., grades, assignment scores, usage logs), nor applying analytical methods such as regression or structural equation modeling. As a result, the statistical analyses remain exploratory and descriptive, substantially limiting the interpretive strength and persuasive power of the findings.
  2. If correctly understood, the survey instrument was developed ad hoc for this study, yet no information is provided regarding its reliability or validity, and no evidence of construct validation is presented. Moreover, the manuscript does not demonstrate how the Likert-scale items are theoretically or empirically aligned with the conceptual structure of engagement, thus weakening measurement adequacy and conceptual clarity.
  3. Although statistically significant gender differences are reported, the manuscript offers insufficient theoretical explanation for why such differences occur. While the authors refer to factors such as the minority status of women in STEM fields and technological self-efficacy, these ideas are not explicitly connected to the theoretical framework or empirically supported within the presented data, creating a risk of speculative interpretation.
  4. The generalizability of the findings is highly limited due to the sample being drawn from a single institution and a single engineering program in Chile. As the authors acknowledge, cultural, institutional, and technological infrastructure factors may substantially influence the topic; yet, without comparative analysis, it is impossible to observe differences across regions, countries, or disciplines.
  5. Although the paper emphasizes an ambivalence model, it does not explain how such ambivalence leads to learning engagement or outcomes through any causal mechanism. The absence of a theoretical or empirical pathway linking ambivalence → engagement → learning outcomes restricts the contribution of the study to a merely exploratory level.
  6. Regarding qualitative data, the qualitative component depends solely on a single data source—open-ended survey responses—resulting in limited depth and contextual richness. Because qualitative data were collected exclusively from textual responses within an online survey rather than through interviews or focus groups, the participants’ narratives tend to be brief and lacking background context. This limits the capacity to explore emotional, motivational, and experiential processes related to AI usage, and weakens the explanatory potential of the ambivalence claim.
  7. The procedure for thematic analysis is described only briefly, providing insufficient evidence of analytic rigor. Although the authors state that three researchers independently coded the data before reaching consensus, key methodological details—such as the development of the coding frame, the structure of themes and subthemes, inter-coder agreement measures, and representative quotations—are not provided, reducing transparency and reproducibility.
  8. The qualitative findings are not adequately integrated with the quantitative results from a mixed-method triangulation perspective. Although the manuscript asserts the use of triangulation, it does not illustrate how quantitative and qualitative findings were meaningfully connected. For example, statistical gender differences are presented, yet the qualitative analysis does not examine corresponding differences in emotional responses or attitudes. This disconnect weakens interpretive coherence and mixed-method validity.
  9. The central concept of ambivalence is not theoretically grounded within the qualitative analysis. Although the coexistence of positive and negative perceptions is described, the manuscript does not elaborate the theoretical definition, constituent dimensions, developmental mechanisms, or distinctions from prior research. Consequently, the interpretation of ambivalence appears more inferential than data-driven.
  10. The absence of direct participant quotations significantly reduces the vividness and credibility (thick description) of the qualitative findings. Since no representative verbatim statements are provided, readers cannot evaluate how themes emerged from the raw data, diminishing the transparency and persuasiveness of the qualitative analysis.
Comments on the Quality of English Language

This study aims to analyze the perceptions of AI tool usage and its impact on student engagement among 170 engineering students at a university in Chile, using a mixed-methods approach. Considering the recent surge of research on the use and effectiveness of AI in education, and the timeliness of the topic, the research focus and purpose may hold a degree of scholarly relevance. However, substantial discussion and improvement are required regarding the following issues.

  1. Although the study claims educational effects of AI usage experience, it lacks quantitative indicators that directly measure learning engagement or academic outcomes. While the concept of engagement is emphasized, the analysis relies solely on perception-based survey items, without incorporating behavioral or performance-based data (e.g., grades, assignment scores, usage logs), nor applying analytical methods such as regression or structural equation modeling. As a result, the statistical analyses remain exploratory and descriptive, substantially limiting the interpretive strength and persuasive power of the findings.
  2. If correctly understood, the survey instrument was developed ad hoc for this study, yet no information is provided regarding its reliability or validity, and no evidence of construct validation is presented. Moreover, the manuscript does not demonstrate how the Likert-scale items are theoretically or empirically aligned with the conceptual structure of engagement, thus weakening measurement adequacy and conceptual clarity.
  3. Although statistically significant gender differences are reported, the manuscript offers insufficient theoretical explanation for why such differences occur. While the authors refer to factors such as the minority status of women in STEM fields and technological self-efficacy, these ideas are not explicitly connected to the theoretical framework or empirically supported within the presented data, creating a risk of speculative interpretation.
  4. The generalizability of the findings is highly limited due to the sample being drawn from a single institution and a single engineering program in Chile. As the authors acknowledge, cultural, institutional, and technological infrastructure factors may substantially influence the topic; yet, without comparative analysis, it is impossible to observe differences across regions, countries, or disciplines.
  5. Although the paper emphasizes an ambivalence model, it does not explain how such ambivalence leads to learning engagement or outcomes through any causal mechanism. The absence of a theoretical or empirical pathway linking ambivalence → engagement → learning outcomes restricts the contribution of the study to a merely exploratory level.
  6. Regarding qualitative data, the qualitative component depends solely on a single data source—open-ended survey responses—resulting in limited depth and contextual richness. Because qualitative data were collected exclusively from textual responses within an online survey rather than through interviews or focus groups, the participants’ narratives tend to be brief and lacking background context. This limits the capacity to explore emotional, motivational, and experiential processes related to AI usage, and weakens the explanatory potential of the ambivalence claim.
  7. The procedure for thematic analysis is described only briefly, providing insufficient evidence of analytic rigor. Although the authors state that three researchers independently coded the data before reaching consensus, key methodological details—such as the development of the coding frame, the structure of themes and subthemes, inter-coder agreement measures, and representative quotations—are not provided, reducing transparency and reproducibility.
  8. The qualitative findings are not adequately integrated with the quantitative results from a mixed-method triangulation perspective. Although the manuscript asserts the use of triangulation, it does not illustrate how quantitative and qualitative findings were meaningfully connected. For example, statistical gender differences are presented, yet the qualitative analysis does not examine corresponding differences in emotional responses or attitudes. This disconnect weakens interpretive coherence and mixed-method validity.
  9. The central concept of ambivalence is not theoretically grounded within the qualitative analysis. Although the coexistence of positive and negative perceptions is described, the manuscript does not elaborate the theoretical definition, constituent dimensions, developmental mechanisms, or distinctions from prior research. Consequently, the interpretation of ambivalence appears more inferential than data-driven.
  10. The absence of direct participant quotations significantly reduces the vividness and credibility (thick description) of the qualitative findings. Since no representative verbatim statements are provided, readers cannot evaluate how themes emerged from the raw data, diminishing the transparency and persuasiveness of the qualitative analysis.

Author Response

Comment 1: This study aims to analyze the perceptions of AI tool usage and its impact on student engagement among 170 engineering students at a university in Chile, using a mixed-methods approach. Considering the recent surge of research on the use and effectiveness of AI in education, and the timeliness of the topic, the research focus and purpose may hold a degree of scholarly relevance. However, substantial discussion and improvement are required regarding the following issues.

Although the study claims educational effects of AI usage experience, it lacks quantitative indicators that directly measure learning engagement or academic outcomes. While the concept of engagement is emphasized, the analysis relies solely on perception-based survey items, without incorporating behavioral or performance-based data (e.g., grades, assignment scores, usage logs), nor applying analytical methods such as regression or structural equation modeling. As a result, the statistical analyses remain exploratory and descriptive, substantially limiting the interpretive strength and persuasive power of the findings.

Response 1: We appreciate the reviewer’s thoughtful observation. We agree that behavioural or performance-based indicators (e.g., academic grades, task scores, or system usage logs) and more advanced analytical techniques could provide additional depth to the study of AI-supported learning engagement. However, incorporating these types of measures falls beyond the methodological scope and objectives of the present research, which was designed as an exploratory, perception-based study aimed at understanding students’ emotional, ethical, and cognitive experiences with AI rather than predicting academic performance. We fully acknowledge this limitation in the manuscript and highlight it as an important avenue for future research.

Comment 2: If correctly understood, the survey instrument was developed ad hoc for this study, yet no information is provided regarding its reliability or validity, and no evidence of construct validation is presented. Moreover, the manuscript does not demonstrate how the Likert-scale items are theoretically or empirically aligned with the conceptual structure of engagement, thus weakening measurement adequacy and conceptual clarity.

Response 2: Thank you for this important observation. We agree that the original version of the manuscript did not provide sufficient information regarding the validity of the survey instrument. In the revised manuscript, we expanded the description of the instrument development process by detailing the expert review and pilot testing. These additions strengthen the methodological transparency of the study and directly address the reviewer’s concern.

Comment 3: Although statistically significant gender differences are reported, the manuscript offers insufficient theoretical explanation for why such differences occur. While the authors refer to factors such as the minority status of women in STEM fields and technological self-efficacy, these ideas are not explicitly connected to the theoretical framework or empirically supported within the presented data, creating a risk of speculative interpretation.

Response 3: We thank the reviewer for this insightful comment. In the revised manuscript, we strengthened the theoretical grounding of the gender-related findings by explicitly integrating evidence from STEM education research on technological self-efficacy, anxiety, and risk aversion among female students. Specifically, we incorporated well-established theoretical and empirical contributions into Section 2.3 to explain how gender differences relate to perceptions of competence, confidence, and technological engagement. These additions clarify that the gender differences observed in the study are consistent with documented patterns in the literature and are theoretically aligned with the psychological and perceptual constructs analysed. This enhancement prevents speculative interpretation and reinforces the conceptual coherence of the manuscript

Comment 4: The generalizability of the findings is highly limited due to the sample being drawn from a single institution and a single engineering program in Chile. As the authors acknowledge, cultural, institutional, and technological infrastructure factors may substantially influence the topic; yet, without comparative analysis, it is impossible to observe differences across regions, countries, or disciplines.

Response 4: We appreciate the reviewer’s observation regarding the limited generalizability of the findings. We fully acknowledge that the sample, drawn from a single institution and engineering program in Chile, restricts the extent to which the results can be extrapolated to other cultural, institutional, or disciplinary contexts. For this reason, this limitation is explicitly stated and discussed in the “Limitations and Directions for Future Research” section of the revised manuscript. As noted there, the present study was designed as an exploratory investigation aimed at understanding students’ perceptions and experiences within a specific educational setting rather than producing cross-context comparative claims. We agree that future research incorporating multi-institutional, international, or interdisciplinary samples would provide valuable comparative insight, and we have highlighted this as an important direction for further work. We thank the reviewer for emphasising this point, which helped us clarify the scope and boundaries of the study.

Comment 5: Although the paper emphasizes an ambivalence model, it does not explain how such ambivalence leads to learning engagement or outcomes through any causal mechanism. The absence of a theoretical or empirical pathway linking ambivalence → engagement → learning outcomes restricts the contribution of the study to a merely exploratory level.

Response 5: We appreciate the reviewer’s thoughtful comment. We agree that establishing causal pathways between ambivalence, learning engagement, and academic outcomes would require a different research design—one incorporating longitudinal, behavioural, or experimental data. However, the present study was intentionally designed as an exploratory investigation centred on students’ perceptions, emotional tensions, and ethical concerns regarding AI, rather than as a causal model of learning outcomes. Our aim was to describe and interpret the coexistence of positive and negative evaluations of AI and how these perceptions relate to students’ sense of confidence, autonomy, and academic decision-making—not to test directional effects.

We have clarified this scope more explicitly in the manuscript and emphasised it both in the introduction and in the Limitations and Directions for Future Research section. We thank the reviewer for highlighting this point, which helped refine the boundaries and aims of the present study.

Comment 6: Regarding qualitative data, the qualitative component depends solely on a single data source—open-ended survey responses—resulting in limited depth and contextual richness. Because qualitative data were collected exclusively from textual responses within an online survey rather than through interviews or focus groups, the participants’ narratives tend to be brief and lacking background context. This limits the capacity to explore emotional, motivational, and experiential processes related to AI usage, and weakens the explanatory potential of the ambivalence claim.

Response 6: Thank you for this thoughtful observation. We agree that open-ended survey responses, while valuable for capturing students’ spontaneous reflections, offer more limited depth and contextual richness compared to qualitative interviews or focus groups. This constraint is inherent to the design of the present study, which was conceived as an exploratory, perception-based investigation aimed at identifying patterns of ambivalence, emotional tensions, and ethical concerns in a broader student population. As such, the qualitative component was intended to complement—not replace—quantitative insights by providing illustrative evidence of the nuances in students’ experiences. We fully acknowledge this limitation and have addressed it explicitly in the Limitations and Directions for Future Research section of the revised manuscript.

Comment 7: The procedure for thematic analysis is described only briefly, providing insufficient evidence of analytic rigor. Although the authors state that three researchers independently coded the data before reaching consensus, key methodological details—such as the development of the coding frame, the structure of themes and subthemes, inter-coder agreement measures, and representative quotations—are not provided, reducing transparency and reproducibility.

Response 7: We appreciate the reviewer’s careful attention to the qualitative methodology. In response, we have substantially expanded the description of the thematic analysis procedure to improve transparency and demonstrate analytic rigour. The revised section now details the development of the coding frame, the steps followed to construct themes and subthemes, and the procedures used for independent coding and consensus-building. We also report the level of intercoder agreement at the initial coding stage and included representative participant quotations to strengthen trustworthiness. These additions provide a clearer and more reproducible account of the analytic process.

Comment 8: The qualitative findings are not adequately integrated with the quantitative results from a mixed-method triangulation perspective. Although the manuscript asserts the use of triangulation, it does not illustrate how quantitative and qualitative findings were meaningfully connected. For example, statistical gender differences are presented, yet the qualitative analysis does not examine corresponding differences in emotional responses or attitudes. This disconnect weakens interpretive coherence and mixed-method validity.

Response 8: We appreciate the reviewer’s careful attention to this methodological issue. We acknowledge that the original version of the manuscript used the term triangulation imprecisely, which may have created conceptual confusion. We apologise for this lack of clarity. The study was designed as an exploratory mixed-methods approach in which qualitative data were intended to complement and deepen the descriptive quantitative patterns, rather than to achieve formal methodological triangulation or convergent validation.

To address this concern, we have carefully revised the Methods section to remove references to triangulation and to accurately describe the integration of methods as exploratory complementarity. The final paragraph of the Data Analysis subsection has been rewritten to clarify that qualitative findings were used to contextualise quantitative trends and provide interpretive depth, consistent with the exploratory scope of the study.

Comment 9: The central concept of ambivalence is not theoretically grounded within the qualitative analysis. Although the coexistence of positive and negative perceptions is described, the manuscript does not elaborate the theoretical definition, constituent dimensions, developmental mechanisms, or distinctions from prior research. Consequently, the interpretation of ambivalence appears more inferential than data-driven.

Response 9: We thank the reviewer for this valuable observation. We agree that the initial version of the manuscript did not sufficiently articulate the theoretical foundations of ambivalence nor clearly demonstrate how the construct emerged from the qualitative data. To address this concern, we substantially strengthened the discussion of ambivalence at both the conceptual and empirical levels.

First, we incorporated a concise yet rigorous theoretical definition of ambivalence, based on established literature that explains its constitutive dimensions and psychological mechanisms, in the Discussion section. This addition clarifies that ambivalence refers to the simultaneous coexistence of positive and negative evaluations toward the same object, and explains how this dual appraisal is particularly relevant in technology-mediated learning contexts.

Second, we revised the qualitative analysis in the Discussion to explicitly show how ambivalence emerges directly from students’ narratives. We highlight how students frequently expressed enthusiasm for AI (e.g., efficiency, autonomy, improved comprehension) while simultaneously reporting concerns related to misinformation, ethical risks, dependence, and anxiety. These dual evaluations often appeared within the same response, demonstrating that ambivalence was grounded in the data rather than inferred externally.

Comment 10: The absence of direct participant quotations significantly reduces the vividness and credibility (thick description) of the qualitative findings. Since no representative verbatim statements are provided, readers cannot evaluate how themes emerged from the raw data, diminishing the transparency and persuasiveness of the qualitative analysis.

Response 10: We thank the reviewer for this valuable observation. In response, we have substantially strengthened the qualitative component by incorporating representative verbatim quotations from the student responses (translated into English for clarity). These excerpts illustrate the key thematic categories related to both benefits and limitations of AI use, thereby enhancing descriptive richness, transparency, and alignment with qualitative research standards.

(THE MODIFICATIONS MADE TO THE ORIGINAL MANUSCRIPT, FOLLOWING THE REVIEWERS' OBSERVATIONS, ARE HIGHLIGHTED IN RED)

Round 2

Reviewer 1 Report

Comments and Suggestions for Authors

Please see attached review notes.

Author Response

Comment 1: Although the addition of the three new research questions (RQ1–RQ3) substantially strengthens the study’s conceptual focus. This foundationally reshapes the theoretical and analytical expectations of the paper, although the body of the manuscript (as currently described) does not fully support the motivational backdrop for these questions.

RQ1: The conclusions discuss students’ ambivalence toward AI (enthusiasm in contrast to mistrust, uncertainty, and integrity concerns), yet the manuscript does not sufficiently operationalise these important and foundational constructs.

RQ2: Explores emotional tensions and ethical concerns, yet in the conclusions, emotional dimensions are presented as secondary themes rather than central constructs.

RQ3: Although the intent is to explore noticable gender differences in perceptions, trust and interaction of AI tools, a more transparent analytical procedures for assessesment would be suggested. Potential gender imbalance and sample characteristics could potentially add cultural bias in perceptions of ethics, especially considering this case, a single sample from a Chilean institution.

Response 1: We sincerely appreciate the reviewer’s thoughtful and constructive feedback, which has contributed substantially to improving the conceptual clarity and methodological transparency of the manuscript. In response to your observations, we have introduced several revisions that strengthen the alignment between the research questions, the analytical procedures, and the empirical evidence presented.

First, we acknowledge that the initial formulation of the research questions could have more clearly reflected the structure of the instrument and the scope of the analyses. To address this, we reformulated the research questions so that each corresponds directly to one of the three sections of the questionnaire: patterns of use and adoption (RQ1), perceived usefulness and gender differences (RQ2), and perceived benefits, risks, difficulties, and emotional responses (RQ3). This reformulation improves conceptual coherence and ensures that the analytical expectations are fully consistent with the data collected.

Second, regarding the operationalisation of ambivalence, emotional tensions, and ethical concerns, we have expanded the descriptions in both the Instrument and Data Analysis sections. Section 3.3 now explains that although the open-ended questions explicitly asked about “benefits” and “difficulties/concerns,” their open formulation allowed students to spontaneously express emotions, ethical concerns, and perceptions of risk. Section 3.4 provides a more explicit account of how these emergent expressions were inductively coded and organised into subthemes, allowing us to treat these dimensions as grounded indicators of ambivalence and affective–ethical tensions. These constructs are now more clearly integrated into the Discussion, where they are interpreted as central components of the student experience with AI tools.

With respect to the analysis of gender differences, we maintained the comparison procedures that were already described in the original manuscript (including the use of Mann–Whitney U tests), but we have clarified more explicitly how these results should be interpreted. In particular, we emphasise that the gender analysis has an exploratory character and is constrained by sample imbalance, which prevented more robust comparisons. The expanded “Limitations and Directions for Future Research” section now explicitly acknowledges that the gender-related patterns reported in the manuscript should be viewed as indicative rather than generalisable, and that they may be influenced by the specific institutional and cultural context of a single Chilean university.

Comment 2: Clarity and definitions are missing concerning foundational elements, meaning that more elaboration is need to aspects such as “AI engagement gap” (p. 1, r. 22; p. 14, r. 581). In addition, the relatively few references used to substantiate the motivation for the study presents an apparent weakness. Prior research could have been mobilised more effectively to articulate the problem space and the contribution of the present work with greater precision and elegance. While brought up, in the statement on p.2 (r. 56–58):

"In this sense, AI has unique potential to personalise learning and offer adaptive support that promotes greater autonomy, which are decisive elements in promoting sustained participation."

This functions as a rhetorical bridge but is insufficiently supported by the cited literature. A suggestion is to promote stronger grounding in prior empirical and theoretical work to improve and more precisely justify the arising need.

Response 2: We sincerely thank the reviewer for this valuable observation. In the revised manuscript, we have thoroughly strengthened the introduction to address the concerns raised.

First, we enhanced the conceptual clarity by explicitly defining and elaborating on the foundational constructs that structure the study. The revised introduction now provides clear definitions of AI literacy, student engagement, and the AI engagement gap, detailing their cognitive, emotional, and behavioural components and the mechanisms through which they influence students’ interactions with AI tools.

Second, we substantially expanded the theoretical and empirical background supporting the motivation for the study. The revised version incorporates a broader and more current body of literature (e.g., Al-Abdullatif & Alsubaie, 2024; Bond et al., 2021; Das & J.V., 2024; Wang et al., 2025), strengthening the justification for the relevance of AI literacy and the role of pedagogical frameworks in shaping technology-mediated engagement.

Third, in response to the reviewer’s request for greater precision in articulating the “AI engagement gap,” we have added a dedicated explanation of the construct and substantiated it with empirical evidence showing how misalignments in competence, trust, emotional comfort, and autonomy support can lead to inconsistent adoption, avoidance behaviours, or surface-level use of AI tools (Zhai et al., 2024; Yang et al., 2025). We further incorporated literature from STEM and educational technology research to document how these dynamics are especially salient in gendered patterns of technology engagement (Beyer, 2014; Cai et al., 2017; Cheryan et al., 2016).

Finally, we improved the overall narrative coherence and elegance of the introduction by reorganising paragraphs, strengthening transitions, and clearly articulating how the identified gaps lead to the three research questions guiding the study.

Comment 3: The reviewer does acknowledge and appreciate the clarifications provided on pp. 6–7 (r. 267–285), where the mixed-methods procedure is described in a more transparent and systematic way.

Response 3: We appreciate the reviewer’s positive feedback regarding the improved description of the mixed-methods procedure. We are pleased that the revisions enhanced transparency and methodological clarity.

Comment 4: In the discussion section, the proposively new deepened version, is more thorough in the sense that it has more references. Despite this promising effort, it is a revision that has opened up for more concerns that merely clarifying past doubts. For instance, the inclusion of Tossell et al. and Jonas C Ziegler introduces two new perspectives; however, these sources are not meaningfully integrated into the theoretical foundation. Instead, they appear as later additions that are discussed without being anchored in the earlier conceptual framing. This raises concerns about whether new elaborating depth and richness is reached through a quick fix, as these new added references do increase the number of references but are not properly anchored in earlier sections.

Response 4: We thank the reviewer for this insightful observation. We fully agree that, in the previous version, the incorporation of Tossell et al. and Jonas & Ziegler might have appeared as late additions, insufficiently anchored in the earlier conceptual development. In response, we substantially revised the theoretical framework and discussion to ensure conceptual coherence and strong alignment between cited literature and the analytic interpretation of results.

Specifically, we integrated the construct of attitudinal ambivalence (Jonas & Ziegler, 2007) directly into Section 2.1, where we now define and contextualise ambivalence as a central lens for understanding students’ mixed evaluations of AI. This conceptual integration allows the discussion to draw on ambivalence as a theoretically grounded, not ad hoc, construct when interpreting students’ simultaneous enthusiasm and mistrust.

Likewise, the contribution of Tossell et al. (2024) is now incorporated into Section 2.2, where it is framed within the broader literature on AI-mediated scaffolding, autonomy support, and step-by-step clarification processes. This framing allows its later use in the discussion to be fully consistent with the motivational and cognitive mechanisms described under the Self-Determination Theory (SDT) lens.  

In revising the discussion, we removed isolated mentions and ensured that all references—including the two highlighted by the reviewer—are interpreted explicitly through the theoretical constructs previously established (ambivalence, psychological-need satisfaction and frustration, perceived usefulness, credibility, and gendered competence beliefs). As a result, the discussion no longer introduces new conceptual perspectives but elaborates the findings strictly within the framework developed in Section 2.

We believe these revisions substantially strengthen theoretical cohesion and ensure that the manuscript advances a more integrated and conceptually grounded interpretation of the results.

Comment 5: Given the short time frame in which the authors report having revised the manuscript, it is notable, and somewhat unexpected, that so many new elements have been introduced. This pattern is also evident in the discussion, where new references (e.g., Deci C Ryan; Ryan C Deci) appear but are not used to develop a coherent or extended account of Self-Determination Theory that would support the argumentative progression on p.13 (r. 509–512). Their late appearance further contributes to the impression of superficial updating rather than deeper conceptual revision.

Response 5: We appreciate the reviewer’s concern regarding the number of revisions introduced within a relatively short time frame. We would like to clarify that the limited time elapsed between submissions does not reflect a lack of dedication or superficial updating. On the contrary, the revision process involved an intensive and systematic reworking of the theoretical framework, with special attention to conceptual coherence and methodological rigor.

In response to the reviewer’s observation about the inclusion of Self-Determination Theory (SDT), we acknowledge that in the previous version SDT appeared only indirectly in the discussion. To address this concern, we have substantially restructured and expanded the theoretical framework to include a coherent, properly anchored exposition of SDT—specifically, the roles of autonomy, competence, and relatedness in shaping motivation and engagement. This conceptual revision ensures that the references to Deci & Ryan and Ryan & Deci are no longer late additions but form an integral part of the analytical foundation of the manuscript.

Comment 6: In summary, it is both surprising and somewhat disappointing that several sections marked in red show minimal substantive development. Rather than addressing earlier structural concerns, these passages have largely been cosmetically updated, with minor details added and occasional new references inserted, yet without resolving the underlying issues. Although the revisions have improved the overall readability of the paper, challenges remains, as the current verison is still not capable to demonstrate the level of conceptual integration and structural improvement required. Despite the manuscript adresses a criticial topic on how to approach AI in engineering education it requires deeper analytical alignment, and improved connection between methodological steps and framed conclusions. By strengthening these elements the paper will significantly improve in both rigour and clarity.

Response 6: We sincerely appreciate the reviewer’s detailed and critical feedback. We understand the concern that some sections in the earlier version may have appeared to undergo only superficial updates. In light of this, we undertook a thorough and substantive revision of the manuscript, focusing not on cosmetic adjustments but on addressing the deeper structural and conceptual issues identified in the initial review.

To accomplish this, we rewrote, restructured, and complemented several central sections to ensure full conceptual alignment across the manuscript. The theoretical framework was significantly strengthened, integrating ambivalence, Self-Determination Theory, and perceptual factors in a coherent and systematic manner rather than as isolated additions. These elements are now fully developed and interconnected in Section 2, providing the conceptual scaffolding required for the later analytical steps. Additionally, we clarified and reinforced the connection between the methodological procedures and the interpretive claims presented in the results and discussion, ensuring consistent alignment between the mixed-methods design and the conclusions derived from it.

The discussion section was also comprehensively revised so that its interpretive arguments closely follow the theoretical foundations established earlier. Rather than introducing new concepts late in the manuscript, the discussion now builds directly and consistently on the framework presented in the revised Sections 2.1–2.3, thereby enhancing the analytical depth, structural clarity, and argumentative coherence of the whole manuscript. We believe these changes address the reviewer’s concerns regarding conceptual integration and structural improvement and substantially increase the manuscript’s rigor and clarity.

(THE MODIFICATIONS MADE TO THE MANUSCRIPT, FOLLOWING THE REVIEWERS' OBSERVATIONS IN THE SECOND ROUND, ARE HIGHLIGHTED IN BLUE)

Reviewer 4 Report

Comments and Suggestions for Authors

Overall, the revised manuscript demonstrates improvements in theoretical grounding, qualitative analysis, and the articulation of methodological limitations. I would, however, suggest two minor additions to further enhance conceptual clarity.

First, it may be useful to clarify that ‘engagement’ in this study reflects perceived engagement-related experiences rather than behavior-based or performance-based engagement, so as to avoid potential ambiguity. Second, because the survey instrument was developed specifically for this exploratory study, explicitly stating that full psychometric validation was beyond the scope of the current research would help align the measurement approach with the study’s aims and prevent misunderstanding regarding the instrument’s status.

Comments on the Quality of English Language

Overall, the revised manuscript demonstrates improvements in theoretical grounding, qualitative analysis, and the articulation of methodological limitations. I would, however, suggest two minor additions to further enhance conceptual clarity.

First, it may be useful to clarify that ‘engagement’ in this study reflects perceived engagement-related experiences rather than behavior-based or performance-based engagement, so as to avoid potential ambiguity. Second, because the survey instrument was developed specifically for this exploratory study, explicitly stating that full psychometric validation was beyond the scope of the current research would help align the measurement approach with the study’s aims and prevent misunderstanding regarding the instrument’s status.

Author Response

Comment 1: First, it may be useful to clarify that ‘engagement’ in this study reflects perceived engagement-related experiences rather than behavior-based or performance-based engagement, so as to avoid potential ambiguity.

Response 1: Thank you for this helpful observation. We agree that clarifying the nature of the engagement construct strengthens the conceptual precision of the manuscript. To address this, we added an explicit statement in the Instrument and Procedure section (Section 3.3) noting that the questionnaire captures perceived engagement-related experiences rather than behavioural or performance-based indicators.

Comment 2: Second, because the survey instrument was developed specifically for this exploratory study, explicitly stating that full psychometric validation was beyond the scope of the current research would help align the measurement approach with the study’s aims and prevent misunderstanding regarding the instrument’s status.

Response 2: Thank you for this valuable suggestion. We agree that clarifying the scope and purpose of the survey instrument will prevent potential misunderstandings regarding its level of validation. In response, we added a statement in the Instrument and Procedure section (Section 3.3) indicating that, given the exploratory nature of the study, the instrument underwent expert review for content validity, but full psychometric validation was beyond the scope of the present research. This clarification appropriately aligns the measurement approach with the study’s exploratory aims.

(THE MODIFICATIONS MADE TO THE MANUSCRIPT, FOLLOWING THE REVIEWERS' OBSERVATIONS IN THE SECOND ROUND, ARE HIGHLIGHTED IN BLUE)